# Energy Balance, Productivity and Resource-Use Efficiency of Diverse Sustainable Intensification Options of Rainfed Lowland Rice Systems under Different Fertility Scenarios

Teekam Singh [1,2], Ram Swaroop Bana [1,*], Bhabani Sankar Satapathy [2,3], Banwari Lal [2,4], Akshay Kumar Yogi [1] and Raj Singh [1]

1    Division of Agronomy, ICAR-Indian Agricultural Research Institute, New Delhi 110012, India; teekam.singh@icar.gov.in (T.S.); akyogi37@gmail.com (A.K.Y.); rajsingh221996@gmail.com (R.S.)
2    Regional Rainfed Lowland Rice Research Station, Gerua, Kamrup 781102, India; bhabani.satapathy@icar.gov.in (B.S.S.); banwari.lal@icar.gov.in (B.L.)
3    ICAR-National Rice Research Institute, Cuttack 753006, India
4    ICAR-Indian Institute of Pulses Research, Regional Centre, Bikaner 334006, India
*    Correspondence: rsbana@gmail.com

**Abstract:** Rice-based cropping systems (RBCS) are a kingpin of global food security and rice fallow is one of the largest (>14 m ha) RBCS. A three-year study was carried out to develop sustainable intensification options and efficient nutrient management protocols of RBCS with greater water and energy productivity and more profits. Rice-lentil, rice-linseed and rice-rapeseed systems were tested in a split-plot design with nutrient management practices involving fertilizer levels (50%, 75% and 100% recommended fertilizer dose; RDF), green manuring with *Sesbania* (SGM) and rice residue incorporation (RRI). The results indicated that SGM produced significantly better rice productivity, enhanced 6.4–22.7% yield of succeeding crops and increased profits by ~20%. Application of 75 or 100% of RDF produced 24.5–30.3% higher grain yield of *rabi* crops. System intensification resulted in an additional rice equivalent yield (REY) of ~1–1.6 t ha$^{-1}$. SGM consumed relatively more energy (76,793 MJ ha$^{-1}$) but at the same time, resulted in higher energy output (182,657 MJ ha$^{-1}$), net energy (105,864 MJ ha$^{-1}$), energy intensity (1.68 MJ INR$^{-1}$) and human energy profitability (787) than the RRI. However, RRI recorded a higher energy ratio (2.42), energy productivity (0.082 kg MJ$^{-1}$) and energy profitability (1.42 kg MJ$^{-1}$). The rice-linseed cropping system resulted in greater system productivity, higher energy output (186,305 MJ ha$^{-1}$) and net energy (112,029 MJ ha$^{-1}$) than other systems. Overall, considering energy productivity, resource-use efficiency and profits, a rice-linseed system coupled with SGM and 75% RDF may be recommended as a sustainable intensification option in RBCS.

**Keywords:** energy efficiency; rice fallows; sustainable intensification; water productivity

## 1. Introduction

Rice (*Oryza sativa* L.) is the staple food for ~50% of the human race and rice-based production systems are a kingpin of global food and nutritional security [1,2]. Among diverse rice-based systems, rice fallow is a major cropping system which covers 14.6 million ha of land, 80% of which is located in India, mostly in the hilly to undulant Eastern Plateau Zone [3]. Almost the entire farming system of this most populous and economically and ecologically fragile part of the world is dependent on rice [3,4]. Due to a dearth of irrigation infrastructure, mono-cropping of rice is predominant and rainfed rice ecologies are principal systems of rice cultivation with ~80% area in the region grown without irrigation during the South-West monsoon season [5]. The region receives >1500 mm annual average precipitation, which is adequate to nurture a short-duration succeeding crop under rice fallows. However, in spite of planned development of Indian agriculture,

>50% of the acreage still remains fallow after rice harvests. To realize greater system productivity, better livelihoods and sustainability under rice-fallow production systems, it is obligatory to explore diverse cropping system intensification options [3]. In some areas of the region, farmers prefer to raise a second crop such as *lathyrus* or lentils without tillage by broadcasting seeds in the standing crop of rice 15–20 days prior to its harvest; the practice is locally known as relay/*utera* cropping. However, the productivity levels of *utera*-crop-based systems are extremely low, mainly owing to adverse soil physical environments, which consequently impede crop growth and nutrients uptake in rice fallows [6]. Therefore, to ensure food and income security of resource-poor masses of the region, there is an urgent need to develop and evaluate sustainable intensification options of rice-fallow system by incorporating a short-duration *rabi* season legume or oilseed crop such as lentil, rapeseed, linseed, field pea, etc., in the system while reducing energy and water footprints.

On the other hand, declining soil fertility and organic carbon in the soils due to no or reduced use of organic manures has emerged as a greater challenge in terms of sustaining productivity of rice and rice-based cropping systems [7]. Nevertheless, synthetic fertilizers use in crop production has assisted in realizing multi-fold productivity enhancements after the green revolution age in South Asia. Simultaneously, injudicious and excessive fertilizer usage has led to several new-generation problems in the farm sector such as deterioration in the soil and groundwater quality, secondary and micronutrients deficiency, environmental pollution and depletion of natural resources [8–11]. Furthermore, during the recent past, growing incidences of rice-stubble burning, mainly owing to rapid expansion of mechanized harvesting of rice and other crops, are jeopardizing the air quality in South Asia, especially in the Indo-Gangetic plains (IGP) and the adjacent tract [9]. Therefore, proficient stubble management has emerged as the foremost challenge for Governments, Scientists and Policymakers.

Incorporation of rice straw, either alone or in combination with green manures, is known to enhance soil health in a holistic manner with increased crop yields [12]. Green manure and crop residues not only supply essential nutrients to the current crop but also leave substantial residual effect on succeeding crops in the rotation [13,14]. Intensive cropping with high-input responsive varieties of crops increased production and productivity but also generated a huge quantity of crop residue in surpluses (e.g., over 400 million tons in India alone) annually, which can be recycled for enriching soil nutrient stock [14]. Unfortunately, the potential advantages in many areas have not been exploited and instead the detrimental practice of largescale stubble-burning is still prevalent. Thus, efficient utilization of crop residues for soil fertility replenishment is the need of the hour in the face of the changing climate scenario. Many scientific studies suggest partial supplementation of nutrient management recommendations by replacement of some quantity of chemical fertilizers with organic sources of nutrients using integrated nutrient management approaches [15–17]. Alternative approaches such as green manures, rice residue incorporation and inclusion of short-duration pulses and oilseeds in rice-based production systems not only enhance system productivity and farm profits but also assist in improving soil nutrient status, promote soil microbial activity and exert positive effects on chemical, physical and biological soil properties [18–20]. Selection of alternative sources of nutrients and methods of their application for sustaining soil fertility mainly depends on their availability, economics and favorable environment for proper decomposing and mineralization. It has been observed that inclusion of oilseeds and pulses in cereal-based crop rotations produced higher and stable net farm income, in spite of higher input costs, across the soil types, which is mostly due to an increase in organic content and nutrient status of soil [21,22].

However, not much scientific information is available concerning the effects of diverse sustainable intensification options of rice-fallow systems on system productivity, energetics and soil health. Likewise, a research gap also exists for green manuring and residue incorporation effects in intensive cropping systems of rainfed rice regions regarding energy-use and water-use efficiency. Therefore, considering the need of sustainable intensification and the knowledge gaps, the present study was carried out to assess the impact of green

manuring, rice residue incorporation and fertility level on the productivity, resource-use efficiency, energetics and sustainability of rice-based cropping systems under shallow lowlands of North-Eastern India. Further, in order to generate information on consumptive water use in the cropping system mode, water- and land-use efficiency under the diverse sustainable intensification options, and production efficiency of various treatments, the study was undertaken.

## 2. Materials and Methods

### 2.1. Experimental Site and Climate

The field experiments were executed at the ICAR-National Rice Research Institute Regional Station, Gerua, Asom, India (28°14′59″ N, 91°33′44″ E, elevation 49 m amsl) during three consecutive years from 2013–2014, 2014–2015 and 2015–2016. The soil of the experimental field was alluvial clay loam with bulk density of 1.21 Mg m$^{-3}$ and slightly acidic nature (pH 6.01). The soil organic carbon was high (1.18%), with medium in available nitrogen (295 kg ha$^{-1}$), available phosphorous (16.7 kg ha$^{-1}$) and available potash (322 kg ha$^{-1}$). In general, >80% of precipitation is received during South-West monsoon (June to September) but rainfall during dry season is meager. The region also received 2–3 early pre-monsoon showers during April and May which allowed us to take green manure crop before rainy season (*kharif*) rice. Shallow water level and water logging lowlands during most of the *kharif* season provide sufficient moisture which can be utilized by cultivating a winter (*rabi*) pulses and oilseeds crops instead of traditional practice of keeping the fields fallow. The rainfall distribution pattern was very erratic and the whole cropping system received average rainfall of 581, 875 and 1565 mm during three years of experimentation, respectively. Out of the total average rainfall, 93%, 60% and 78% rainfall were received during *kharif* season rice from June to November of all three years, respectively. It has been observed that during the wet season rainfall always exceeded evaporation, while in the dry season the reverse was the case. Maximum temperature varied from 22 to 35.4 °C (average 29.7 °C) whereas minimum temperature ranged from 7.6 to 21 °C (average 14.5 °C) during three years (Figure 1). Average bright sunshine hours were 6.1, 5.4 and 5.2 h, and daily average evaporation rate was 2.82, 3.01 and 3.32 mm during 2013–2014, 2014–2015 and 2015–2016, respectively.

### 2.2. Experiment Details

The experiments were carried out in split plot design with three replications during *kharif* rice with a plot size of 54 m$^2$ (9 m × 6 m) whereas *rabi* crops were accommodated in sub-sub plots 18 m$^2$ (6 m × 3 m) in split-split plot design. There were 24 treatment combinations *viz.*, *Sesbania* green manuring and rice residue incorporation in main plots and four fertility levels (control, 50%, 75% and 100% recommended doses of fertilizer (RDF) in sub plots for rice and three succeeding *rabi* crops (lentil, linseed and rapeseed) in sub-sub plots. Based on the 100% RDF (80:40:40 kg ha$^{-1}$ N-P$_2$O$_5$-K$_2$O), other fertility levels were calculated and applied to the respective plots. Green manure crop of *Sesbania aculeata* was buried two weeks prior to transplanting; however, rice residue 5 t ha$^{-1}$ was buried in the soil one month before transplanting during the last week of June every year. In *kharif* season, rice variety Naveen was transplanted in the last week of July during all three years. In the *rabi* season, succeeding lentil, linseed and rapeseed were sown in the second week of November in every year.

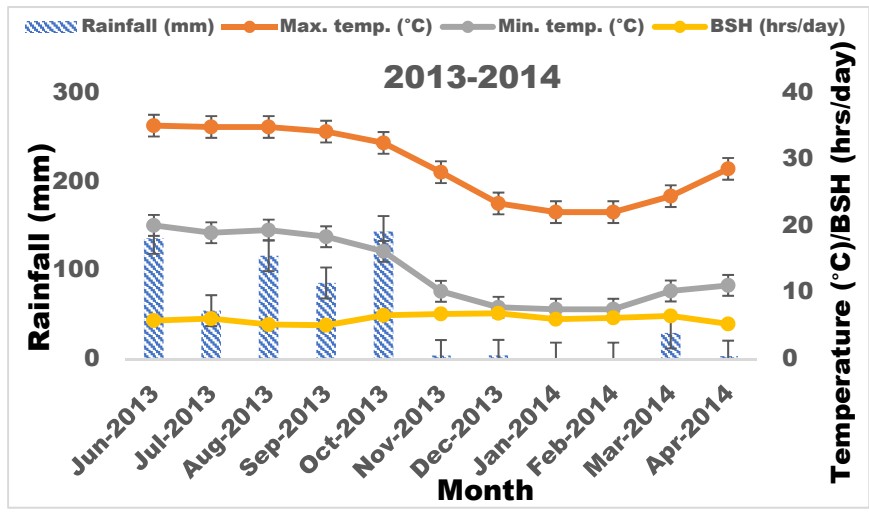

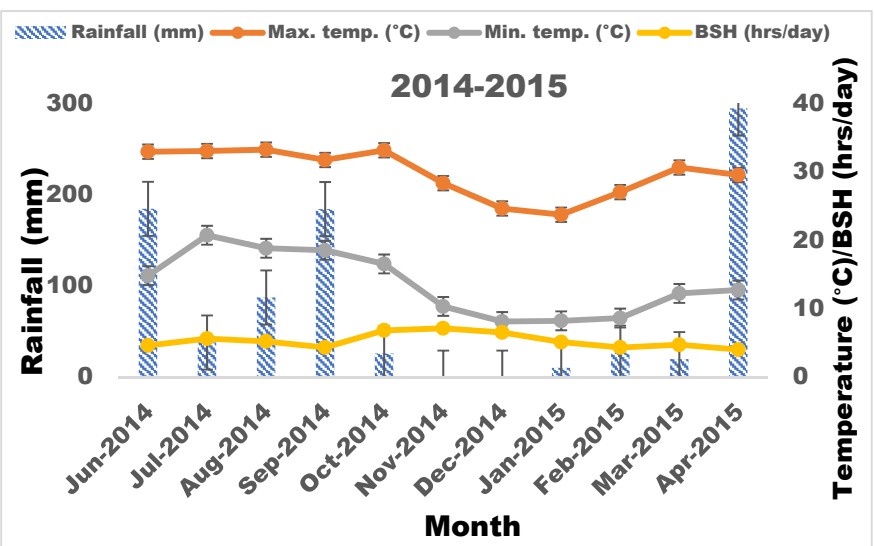

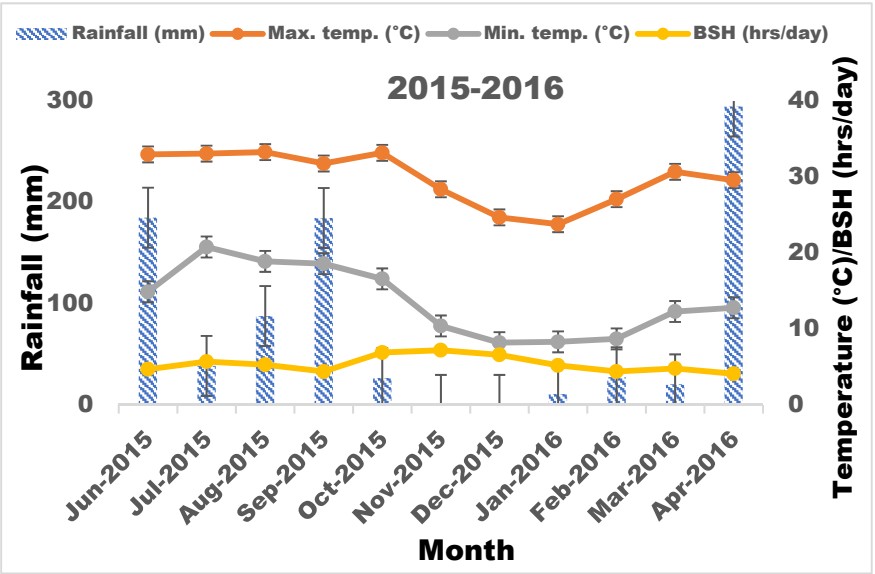

**Figure 1.** Weather parameters during cropping system cycle in three growing seasons 2013–2014, 2014–2015 and 2015–2016 at Experimental site Gerua, Asom, India.

*2.3. Crop Management Practices*

The details of agronomic management practices presented in Table 1. After green manuring and rice residue incorporation 2 weeks and 1 month before transplanting, respectively; field was prepared with 3 tractor drawn ploughings using 9-tyned cultivator and disc-harrow followed by puddling and field leveling during *kharif* season for transplanting of rice. Rice was transplanted at 20 × 10 cm spacing using 25 days old seedlings. The NPK nutrients were applied through urea, diammonium phosphate (DAP) and muriate of potash (MOP) to the *kharif* rice as per treatments. Full dose of phosphorous, one-third of N and three-quarters of potash were applied as basal at the time of transplanting. Remaining N was applied in two equal splits at maximum tillering and panicle initiation stage, whereas one-quarter of potash was applied as top dressing at panicle initiation. After harvesting of rice, residues were removed, and field was prepared by two cross dry ploughing using 9-tyned cultivator and planking by a wooden plank. Dry season crops were manually sown on residual fertility and moisture as no fertilizers and irrigations were applied to the *rabi* crops. Plant protection measures were not required throughout the investigation, except pre-emergence herbicide pretilachlor at 0.75 kg a.i. ha$^{-1}$ which was used to control weeds in rice.

**Table 1.** Management practices for raising individual crops during the field experimentation.

| Crop | Variety | Seed Rate (kg ha$^{-1}$) | Crop Season | Nutrient Applied (kg ha$^{-1}$) | Time of Application of Fertilizers | Weeding/Plant Protection |
|---|---|---|---|---|---|---|
| *Sesbania* green manure (GM) | Local seed | 50.0 | Pre *kharif* | - | GM crop buried two weeks before transplanting | - |
| Rice residue incorporation (RRI) | - | - | - | - | RRI one month prior to transplanting 5 t ha$^{-1}$ | - |
| Rice | Naveen | 35.0 | *Kharif* | 80:40:40 (100% RDF) | 1/3 N + full P + 2/3 K as basal, 1/3 N at active tillering and 1/3 N + 1/3 K at panicle initiation | Pre-emergence application of Pretilachlor 1.0 L a.i. ha$^{-1}$ |
| Lentil | PL 406 | 40.0 | *Rabi* | Grown on residual nutrients of previous crop | | |
| Linseed | T 397 | 15.0 | *Rabi* | Grown on residual nutrients of previous crop | | |
| Rapeseed | TS 36 | 5.0 | *Rabi* | Grown on residual nutrients of previous crop | | |

*2.4. Biometric Observations, Yield and Economics*

Randomly, ten representative panicles were harvested for each plot at maturity to record biometric observations (panicle weight and length, filled and unfilled grains per panicle). Filled grains were separated by submerging threshed grains in the normal tap water. Fertility percentage was calculated as the number of submerged grains divided by the total spikelet number. One thousand seeds were counted and weighed to record 1000-grain weight. One square meter quadrat was used to harvest 34 plants (hills) in the middle of each plot, to determine yield and yield components. After the total panicles were counted, all the spikelets were threshed out from panicles, weighed to determine grain yield at 14% moisture content. Straw was kept for open sun drying until its weight reached constant at 14% moisture and was weighed to calculate straw yield. Similarly, plant stand, plant height, straw and seed yield of *rabi* season crops were recorded from one square meter quadrat area.

Rice equivalent yield (REY) was calculated to compare system performance by converting the yield of oilseeds and pulses into an equivalent rice yield based on Government minimum support price, using the formula:

$$REY = Yx(Px/Pr)$$

where, $Yx$ is the yield of *rabi* crops (kg ha$^{-1}$), $Px$ is the price of *rabi* crops (INR [Indian rupee] kg$^{-1}$), and $Pr$ is the price of rice (INR kg$^{-1}$). Prices of individual inputs and outputs were assumed to be stable during the experimental period.Net returns (INR ha$^{-1}$) and benefit:

cost (B:C) ratio were calculated by considering the minimum support price fixed by the government for paddy grains (INR 13.6 kg$^{-1}$) and cost of cultivation. Net returns, B:C ratio, production efficiency and economic efficiency were calculated by the following formulas as suggested by Lal et al. 2017:

$$Net\ returns\ (INR\ ha^{-1}) = Gross\ return\ (INR\ ha^{-1}) - Cost\ of\ cultivation$$
$$B : C\ ratio = Gross\ returns\ (INR\ ha^{-1})/cost\ of\ cultivation\ (INR\ ha^{-1})$$
$$Production\ efficiency\ (kg\ ha^{-1}day^{-1}) = grain\ yield\ (kg\ ha^{-1})/total\ duration\ of\ crops\ (days)$$
$$Economic\ efficiency\ (INR\ ha^{-1}day^{-1}) = Net\ returns\ (INR\ ha^{-1})/total\ duration\ of\ crops\ (days)$$

### 2.5. Input–Output Energy Analysis and Sustainability

Energy input–output flow of the cropping systems was estimated using crop management practices (seed, fertilizers, weeding, machinery operations and manual labor and other inputs used (Table 2) and grain and straw yields recorded. Inputs and outputs were converted from physical to energy unit measures through published conversion coefficients given in Table 3 [23–25]. Inputs energy equivalents were averaged out to obtain an estimate of total energy inputs used in production. Output energy was calculated by multiplying economic grain yield and straw with its corresponding energy equivalents. Net energy return is the difference between the output energy produced and the total energy required in terms of inputs [24,25].

**Table 2.** Inputs consumption in various rice-based cropping systems under sesbania green manuring and rice residue incorporation in rainfed lowlands.

| Input | *Sesbania* Green Manuring | | | | Rice Residue Incorporation | | | |
|---|---|---|---|---|---|---|---|---|
| | Rice Fallow | Rice Lentil | Rice Linseed | Rice Rapeseed | Rice Fallow | Rice Lentil | Rice Linseed | Rice Rapeseed |
| Seed of green manure crop (kg ha$^{-1}$) | 50 | 50 | 50 | 50 | | | | |
| Green manure matter/rice residue (t ha$^{-1}$) | 5.4 | 5.4 | 5.4 | 5.4 | 5 | 5 | 5 | 5 |
| Nitrogen (100% N) (kg ha$^{-1}$) | 80 | 80 | 80 | 80 | 80 | 80 | 80 | 80 |
| Phosphorus (kg ha$^{-1}$) | 40 | 40 | 40 | 40 | 40 | 40 | 40 | 40 |
| Potassium (kg ha$^{-1}$) | 40 | 40 | 40 | 40 | 40 | 40 | 40 | 40 |
| Seed (kg ha$^{-1}$) | 35 | 75 | 75 | 40 | 35 | 75 | 75 | 40 |
| Pesticides (kg ha$^{-1}$) | 1 | 1 | 1 | 2 | 1 | 1 | 1 | 2 |
| Diesel (L ha$^{-1}$) | 35 | 47 | 47 | 47 | 35 | 47 | 47 | 47 |
| Tractor (h ha$^{-1}$) | 12 | 18 | 18 | 18 | 12 | 18 | 18 | 18 |
| Human labor (8 h d$^{-1}$ ha$^{-1}$) | | | | | | | | |
| Men | 105 | 135 | 139 | 145 | 105 | 135 | 139 | 145 |
| Women | 82 | 102 | 108 | 110 | 82 | 102 | 108 | 110 |

**Table 3.** Energy equivalents for inputs and outputs from agricultural production [23–25].

| Particulars | Unit | Energy Equivalent (MJ Unit$^{-1}$) |
|---|---|---|
| **A.　Inputs** | | |
| Seed of green manure crop | kg | 14.70 |
| Green manure dry matter | kg | 12.5 |
| Rice residue incorporation | kg | 12.50 |
| Nitrogen (N) | kg | 66.14 |
| Phosphorus (P$_2$O$_5$) | kg | 12.44 |
| Potassium (K$_2$O) | kg | 11.15 |
| Seed | kg | 14.70 |
| Herbicides | kg | 238.00 |
| Diesel | L | 56.31 |
| Farm Machinery (Tractor) | h | 62.70 |
| Men labor | h | 1.96 |
| Women labor | h | 1.57 |
| **B.　Output** | | |
| Seed yield of rice, lentil, linseed and rapeseed | kg | 17.00 |
| Straw | kg | 12.50 |

The energy ratio (*ER*) was estimated as

$$ER = \frac{Output\ energy\ \left(MJ\ ha^{-1}\right)}{Input\ energy\ \left(MJ\ ha^{-1}\right)}$$

The energy profitability (*PE*) was estimated as

$$PE = \frac{Net\ energy\ return\ \left(MJ\ ha^{-1}\right)}{Input\ energy\ \left(MJ\ ha^{-1}\right)}$$

Human energy profitability (*HPE*) was determined as

$$HPE = \frac{Output\ energy\ \left(MJ\ ha^{-1}\right)}{Labour\ energy\ \left(MJ\ ha^{-1}\right)}$$

Energy productivity (*EP*) was determined as

$$EP = \frac{Crop\ economic\ yield\ \left(kg\ ha^{-1}\right)}{Energy\ Input\ \left(MJ\ ha^{-1}\right)}$$

Energy intensity (*EI*) was estimated as

$$EI = \frac{Energy\ input\ \left(MJ\ ha^{-1}\right)}{Cost\ of\ production\ \left(INR\ ha^{-1}\right)}$$

Sustainable yield index approach is used to evaluate the cropping system to achieve minimum yield to maintain sustainability. Sustainability yield index (SYI) was calculated as suggested by [26].

$$Sustainability\ yield\ index = \frac{Mean\ yield - Standard\ deviation}{Maximum\ yield}$$

Land-use efficiency (*LUE*) was calculated using the following method [25] and it refers to the extent of land area used in a year.

$$LUE\ (\%) = \frac{TND(i) \times 100}{365}$$

where, *TND*(*i*) denotes No. of days field remained occupied under different crops (*i* = 1 n).

### 2.6. Statistical Analysis

Two-way analysis of variance was used to compare treatment effects on each parameter. The statistical analysis was performed for each parameter studied based on a split-plot and split-split plot design using IASRI online data analysis software (https://iasri.icar.gov.in/ accessed on 19 January 2021). Means were compared with Tukey's honest significant difference (HSD) at the 5% level of significance.

## 3. Results and Discussion

### 3.1. Effect on Growth, Yield Attributes and Productivity of Rice

Growth, yield attributes and grain yield of rice were greatly influenced by green manuring, rice residue incorporation and fertility levels during *kharif* season (Table 4). Green manuring of *Sesbania aculeata* mostly enhanced the productivity of rice through increasing the plant height, panicle length and filled grains per panicle significantly (*p* < 0.05) compared to rice residue incorporation. Some growth and yield attributes such as tillers (m$^2$), total spikelets per panicle, panicle weight, 1000-grain weight and straw yield remained at par under green manuring and rice residue incorporation; however, higher values for these parameters were obtained with green manure which indicated that green manuring is

the better option under shallow lowlands. This mainly happened due to the higher amount of nutrient recycling from the green manure crop than the rice residue incorporation. In situ incorporation of green manure *Sesbania aculeata* led to more recycling of NPK nutrients, which resulted in significantly higher productivity of rice [27]. Green manure and rice straw incorporation stimulated microbial growth and soil microbial community, which enhanced enzymatic activity, and the availability of more nutrients resulted in higher productivity of rice [28,29].

**Table 4.** Growth, yield attributes and productivity of rice as influenced by green manuring, rice residue incorporation and fertility levels under rice-based cropping system. (Mean of 3 years).

| Treatment | Plant Height (cm) | Tillers (m$^{-2}$) | Spikelets Panicle$^{-1}$ | Panicle Length (cm) | Panicle Weight (g) | Filled Grains Panicle$^{-1}$ | Fertility (%) | Test Weight (g) | Straw Yield (kg ha$^{-1}$) | Grain Yield (kg ha$^{-1}$) |
|---|---|---|---|---|---|---|---|---|---|---|
| | | | | **Green manuring and rice residue management** | | | | | | |
| Green manuring | 124.1 [A] | 269.2 | 146.7 | 25.9 [A] | 3.94 | 146.2 [A] | 89.2 | 19.84 | 6086.9 | 5212.6 [A] |
| Rice residue incorporation | 118.1 [B] | 252.4 | 135.0 | 24.9 [B] | 3.72 | 133.6 [B] | 87.4 | 19.79 | 5714.4 | 4896.5 [B] |
| Tukey's HSD at 5% | 1.44 | NS | NS | 0.42 | NS | 7.20 | NS | NS | NS | 104.89 |
| | | | | **Fertility levels** | | | | | | |
| Control | 118.3 | 242.4 [C] | 123.5 [C] | 24.1 [C] | 2.81 [C] | 120.2 [C] | 86.4 | 19.78 | 5313.3 [C] | 4464.8 [C] |
| 50% RDF | 120.5 | 254.4 [BC] | 141.8 [B] | 25.3 [B] | 3.58 [BC] | 139.1 [B] | 88.2 | 19.70 | 5718.3 [BC] | 4980.1 [B] |
| 75% RDF | 122.8 | 269.7 [AB] | 145.8 [AB] | 26.1 [AB] | 4.35 [AB] | 148.3 [AB] | 89.2 | 19.85 | 6146.1 [AB] | 5309.6 [AB] |
| 100% RDF | 122.8 | 276.8 [A] | 152.3 [A] | 26.3 [A] | 4.58 [A] | 152.1 [A] | 89.4 | 19.94 | 6425.0 [A] | 5463.6 [A] |
| Tukey's HSD at 5% | NS | 16.69 | 9.81 | 0.80 | 0.92 | 9.75 | NS | NS | 694.99 | 368.57 |

Note: These letters were used to compare treatments which described in text.

Yield attributes *viz.*, tillers (m$^2$), spikelets per panicle, filled grains per panicle, panicle length and weight were mainly affected by fertility levels; however, plant height, fertility percentage and 1000-grain weight remained unaffected. The higher values of tillers, filled grains per panicle, total spikelets per panicle, panicle length and weight resulted in significantly higher grain and straw yield with 100% recommended dose of fertilizer (RDF) over control and 50% RDF. However, 100% RDF remained statistically at par with 75% RDF under shallow lowland conditions. This might have happened due to a substantial supply of nutrients from green manuring and rice residue incorporation. Thus, 75% RDF along with green manuring or rice residue was sufficient to harvest higher rice productivity in shallow lowlands conditions. Many research findings indicated that some quantity of chemical fertilizers could be replaced with crop residue incorporation and green manuring [27,30,31].

*3.2. Residual Effect on Succeeding Rabi Crops*

Succeeding *rabi* season crops showed significantly higher productivity with green manure compared to rice residue incorporation (Table 5). Increased productivity of succeeding pulses and oilseeds with green manure was mainly due to vigorous growth in terms of plant height. The plant stand (m$^2$) and straw yield of *rabi* crops remained unaffected under both green manure and rice residue incorporation; however, the higher values for plant stand (m$^2$) and straw yield were obtained with green manure. These results further indicated that more photosynthates were mobilized from source to sink with green manure. Green manure from legumes could maintain the microorganisms and microbial enzymatic activity for a longer duration and increased the soil N supply to subsequent crops resulted in higher grain yield [27–29].

**Table 5.** Residual effect of green manuring and fertility levels on growth and yield of succeeding crops. (Mean of 3 years).

| Treatment | Plant Stand (m$^{-2}$) | Plant Height (cm) | Straw Yield (kg/ha) | Grain Yield (kg/ha) |
|---|---|---|---|---|
| **Green manuring and rice residue management** | | | | |
| Green manuring | 147.63 | 52.73 A | 819 | 382 A |
| Rice residue incorporation | 127.55 | 51.32 B | 768 | 336 B |
| Tukey's HSD at 5% | NS | 0.51 | NS | 21.0 |
| **Fertility levels** | | | | |
| Control | 122.92 B | 51.37 | 703 C | 307 C |
| 50% RDF | 133.52 B | 51.80 | 760 BC | 349 B |
| 75% RDF | 146.91 A | 51.88 | 831 AB | 382 A |
| 100% RDF | 147.02 A | 53.05 | 879 A | 400 A |
| Tukey's HSD at 5% | 12.07 | NS | 106.7 | 31.5 |
| **Succeeding Rabi crops** | | | | |
| Lentil | 109.25 B | 29.24 C | 582 C | 375 A |
| Linseed | 202.04 A | 50.73 B | 1051 A | 395 A |
| Rapeseed | 101.48 B | 76.11 A | 746 B | 309 B |
| Tukey's HSD at 5% | 8.11 | 1.05 | 80.5 | 27.4 |

Note: These letters were used to compare treatments which described in text.

The study revealed that fertility levels had the significant effect ($p < 0.05$) on plat stand (m$^2$), straw and seed yield of succeeding *rabi* pulses and oil seeds in rice fallows during post rainy season (Table 5). The seed yield of *rabi* crops with residual fertility of 75 and 100% RDF was significantly better over control (24.5–30.3%) and 50% RDF (9.7–14.7%), respectively. However, the maximum seed and straw yield of *rabi* crops was obtained with residual fertility of 100% RDF but remained statistically at par with 75% RDF residual fertility. Among the succeeding *rabi* crops linseed yielded the maximum productivity, followed by lentil, while rapeseed resulted in the lowest seed yield by a significant amount.

Study showed that the interactive effect was significant ($p < 0.05$) between green manuring and rice residue incorporation on seed yields of succeeding *rabi* crops grown in rice fallows during post-rainy season (Table 6). Lentil and linseed performed significantly ($p < 0.05$) better under green manuring compared to rice residue incorporation; however, succeeding rapeseed productivity remained at par with both green manure and rice residue incorporation. The mean results of 3 years shown that green manuring before rice increased the seed yield of succeeding lentil by 22.7%, followed by linseeds by 11.4% and rapeseed 6.4% over rice residue incorporation. This is mainly due to higher availability of nutrients to succeeding crops under green manure crops [27–29].

**Table 6.** Interaction effect of green manuring and rice residue incorporation on succeeding *rabi* crops productivity (kg ha$^{-1}$). (Mean of 3 years).

| Succeeding *Rabi* Crops | Green Manuring | Rice Residue Incorporation |
|---|---|---|
| Lentil | 413 A | 336 CD |
| Linseed | 416 A | 374 B |
| Rapeseed | 318 D | 299 DE |
| Tukey's HSD at 5% | GM and RRI means at same or different succeeding crop | 38.8 |
| | Succeeding crop means at same or different GM and RRI | 37.1 |

Note: These letters were used to compare treatments which described in text.

### 3.3. Effect of Inclusion of Pulses and Oilseeds in Rice Fallows on System Productivity

The system productivity in term rice equivalent yield (REY) of rice-based cropping systems influenced significantly by green manure, rice residue incorporation and fertility levels (Tables 7–9). On average all rice-based cropping systems had significantly higher REY with green manuring compared to rice residue incorporation. Green manuring helped

to increase the REY of rice-based cropping systems to the tune of 928 to 1648 kg ha$^{-1}$ over rice fallow, whereas REY increased 894 to 1570 kg ha$^{-1}$ with rice residue incorporation. Rice linseed recorded the highest REY under both green manure and rice residue incorporation, followed by rice lentil. However, the REY of rice rapeseed was not affected much under green manure and rice residue incorporation.

**Table 7.** Interaction effect of green manuring and rice residue incorporation on system productivity (kg ha$^{-1}$ REY) of different rice-based cropping systems. (Mean of 3 years).

| Cropping System | Green Manuring | Rice Residue Incorporation |
|---|---|---|
| Rice Fallow | 5213 | 4896 |
| Rice Lentil | 6339 | 5790 |
| Rice Linseed | 6861 | 6466 |
| Rice Rapeseed | 6141 | 5932 |
| Tukey's HSD at 5% | GM and RRI means at same or different cropping system | 261.4 |
| | Cropping system means at same or different GM and RRI condition | 221.1 |

REY = Rice equivalent yield; GM = green manuring; RRI = rice residue incorporation.

**Table 8.** Interaction effect of fertility levels on system productivity (kg ha$^{-1}$ REY) of different rice-based cropping systems. (Mean of 3 years.)

| Cropping System Fertility Level | Cropping Systems | | | |
|---|---|---|---|---|
| | Rice Fallow | Rice Lentil | Rice Linseed | Rice Rapeseed |
| Control | 4465 | 5529 | 6179 | 5408 |
| 50% RDF | 4980 | 6007 | 6431 | 6119 |
| 75% RDF | 5310 | 6157 | 6932 | 6410 |
| 100% RDF | 5464 | 6565 | 7111 | 6209 |
| Tukey's HSD at 5% | Fertility means at same or different cropping system | | | 393.4 |
| | Cropping system means at same or different fertility level | | | 312.7 |

**Table 9.** Effect of green manuring and fertility levels on system productivity and profitability of rice-based cropping system. (Mean of 3 years).

| Treatment | Rice Equivalent Yield (kg ha$^{-1}$) | Production Efficiency (kg day$^{-1}$ ha$^{-1}$) | Net Return (INR ha$^{-1}$) | B:C Ratio |
|---|---|---|---|---|
| **Green manuring and rice residue management** | | | | |
| Green manuring | 6138 [A] | 28.58 | 43,181 [A] | 1.94 [A] |
| Rice residue incorporation | 5771 [B] | 27.22 | 36,185 [B] | 1.76 [B] |
| Tukey's HSD at 5% | 188.2 | NS | 2669 | 0.05 |
| **Fertility levels** | | | | |
| Control | 5395 [C] | 25.14 [B] | 32,701 [C] | 1.73 [C] |
| 50% RDF | 5884 [B] | 28.09 [A] | 38,962 [B] | 1.84 [B] |
| 75% RDF | 6202 [A] | 28.82 [A] | 42,610 [AB] | 1.90 [AB] |
| 100% RDF | 6337 [A] | 29.55 [A] | 44,460 [A] | 1.93 [A] |
| Tukey's HSD at 5% | 286.1 | 2.23 | 4022 | 0.09 |
| **Succeeding Rabi crops** | | | | |
| Rice Fallow | 5055 [C] | 24.76 [D] | 32,217 [C] | 1.78 [C] |
| Rice Lentil | 6065 [B] | 27.65 [C] | 37,818 [B] | 1.77 [C] |
| Rice Linseed | 6663 [A] | 30.13 [A] | 44,762 [A] | 1.90 [B] |
| Rice Rapeseed | 6037 [B] | 29.06 [A] | 43,936 [A] | 1.96 [A] |
| Tukey's HSD at 5% | 156.4 | 1.35 | 2226 | 0.05 |

Note: These letters were used to compare treatments which described in text.

Residual fertility and moisture can be utilized efficiently with inclusion of pulses and oilseeds in rice fallow lands, which also improve system productivity and maintain

sustainability [32]. System productivity in terms of REY of rice linseed was the maximum with 100% RDF and significant over control and 50% RDF but remained at par 75% RDF. The rice-lentil cropping system recorded the maximum REY with 100% RDF and found it to be significantly greater than control: 50% and 75% RDF. However, REY of rice rapeseed was found to be significantly higher with all three fertility levels over control but remained at par with each other, which indicated that rapeseed can be grown at the lowest fertility level, i.e., 50% while growing linseed required 75% RDF and lentil needs 100% RDF in rice. Inclusion of oilseeds and pulses in rice fallows could increase the system productivity by over 2600 kg ha$^{-1}$ with 100% RDF in rice over control. Overall, green manuring registered the significantly higher REY over rice residue incorporation; however, 75 and 100% fertility levels were found to be superior over control and 50% RDF but remained at par with each other (Table 9). Among, cropping systems, rice rapeseed can be grown at the lowest fertility level, i.e., 50% while growing linseed required 75% RDF and lentil needs 100% RDF. Rice linseed recorded the significantly highest system productivity over all other rice-based cropping systems. Many findings indicated that inclusion of oilseeds and pulses in rice fallows is the best strategy for efficient utilization of natural resources such as land, residual moisture and fertility [11,20,32].

### 3.4. Effect of Inclusion of Pulses and Oilseeds in Rice Fallows on Production Efficiency and Profitability

Mean data of the 3-year study revealed that the system production efficiency and profitability in terms of net returns and B:C ratio were significantly higher with green manured fields over rice residue incorporation (Table 9). Inclusion of green manuring in a rice-based cropping system helped to increase net profit by approximately 20% over rice residue incorporation, which might be due to inherent balance availability as well as the nutrients applied to the crop plants. The higher productivity and profitability with green manuring in rice-based cropping system were also reported by [33]. Among the fertility levels, 100% RDF recorded the highest production efficiency, net returns and B:C; however, it remained statistically at par with 75% RDF. Thus, with green manuring or rice residue incorporation in rice-based cropping systems, 25% of fertilizers could be saved without compromising with production and profitability. Ref. [34] also reported that integrated nutrient management with green manuring helps to curtail the fertilizer cost up to ~50% without any reduction in productivity and profitability of the rice-wheat cropping system.

Inclusion of pulses and oilseeds in the rice fallow resulted in the additional REY of around 1000 kg ha$^{-1}$, irrespective of the dry season crop, and it enhanced the productivity of overall system from 5000 kg ha$^{-1}$ to around 6660 kg ha$^{-1}$, in general. Linseed added around 1600 kg ha$^{-1}$, while lentil and rapeseed mustard added around 1000 kg ha$^{-1}$ REY to the system leading to system productivity of approximately 6000 to 6600 kg ha$^{-1}$, which was only 5000 kg ha$^{-1}$ in case of rice fallow. Many studies suggested that inclusion of pulses and oilseeds in rice fallow enhanced the crop productivity by 20–35.5% over the conventional fallow system [25,35,36]. Inclusion of short-duration crops under rainfed conditions is a viable alternative which also opens prospects for expansion in rice-fallow systems [7].

### 3.5. Water-Use, Land-Use Efficiency and Sustainability

The water-use efficiency of the rice-based system with sesbania green manuring was higher than rice residue incorporation (Table 10). Greater water-use efficiency and utilization by crops with green manuring in this experiment was due to efficient soil moisture conservation and higher productivity of rice and subsequent *rabi* crops. Green manures hasten the microbial process and release various organic products which make nutrients available to the crops [37] with higher soil moisture retention capacities and less runoff [38].

**Table 10.** Effect of green manuring and fertility levels on water consumptive use and water use efficiency of rice-based cropping system.

| Treatment | Consumptive Water Use of Rabi Crops (mm ha$^{-1}$) | System Consumptive Water Use (mm ha$^{-1}$) | Water Use Efficiency (kg grain ha$^{-1}$ mm) | Land Use Efficiency (%) | Sustainability Yield Index |
|---|---|---|---|---|---|
| **Green manuring and rice residue management** | | | | | |
| Green manuring | 295.9 | 1331.4 | 4.84 | - | - |
| Rice residue incorporation | 252.0 | 1287.5 | 4.71 | - | - |
| **Fertility levels** | | | | | |
| Control | 274.0 | 1309.5 | 4.36 | - | - |
| 50% RDF | 272.4 | 1307.9 | 4.73 | - | - |
| 75% RDF | 286.4 | 1321.9 | 4.92 | - | - |
| 100% RDF | 263.0 | 1298.5 | 5.11 | - | - |
| **Succeeding Rabi crops** | | | | | |
| Rice Fallow | - | 1035.5 | 4.88 | 32.88 | 0.80 |
| Rice Lentil | 272.4 | 1307.9 | 4.63 | 61.64 | 0.82 |
| Rice Linseed | 270.4 | 1305.9 | 5.10 | 64.38 | 0.84 |
| Rice Rapeseed | 279.0 | 1314.5 | 4.59 | 56.16 | 0.86 |

A fertility level of 100% RDF resulted in maximum system water-use efficiency compared to the others, even though all fertility treatments showed almost similar trends in consumptive use of water. However, water-use efficiency with 75% and 100% RDF was statistically identical. The consumptive water use of all succeeding crops after rice was almost the same due to the growth period but differed in productivity, which resulted in variation in system water-use efficiency. The rice-linseed cropping system recorded the highest water-use efficiency among the rice-based cropping systems, which showed that linseed is more productive in rice fallows under limited soil moisture supply. Water productivity of rice in rice fallow was also low compared to the rice-linseed cropping system.

Land use efficiency and sustainability of rice fallows could be almost doubled with the inclusion of oilseeds and pulses (Table 10). In this experiment, rice linseed recorded higher land-use efficiency (64.4%) followed by rice lentil (61.6%). The sustainability yield index was higher with rice rapeseed followed by the rice-linseed system. Enhancement in land-use efficiency and sustainable yield index due to inclusion of legumes and oilseed crops in the rotation have been reported by several workers in diverse systems [14,25,39].

*3.6. Energy Budgeting*

The energy input requirement of rice with green manuring and rice residue incorporation was recorded as very high (76,793 and 70,720 MJ ha$^{-1}$, respectively). Results also revealed that inclusion of pulses and oilseeds in rice fallows needed only 2344 to 2501 MJ ha$^{-1}$ total energy requirement, which showed that growing these crops in rice fallows is less energy intensive under residual moisture and fertility from green manure or rice residue incorporation. Many earlier findings also reported that pulses and oilseeds were the best-suited crops in terms of energy requirement points of view under rainfed shallow lowlands areas [24,25,39]. Out of total input required for different operations, green manuring and rice residue incorporation consumed the maximum energy (67,500 and 62,500 MJ ha$^{-1}$, respectively) due to their bulky nature and requirement in large quantities. The energy consumption for fertilizers application (6235 MJ ha$^{-1}$), diesel for machinery operations (3513 MJ ha$^{-1}$) and human labor (3316 MJ ha$^{-1}$) stood second, third and fourth, respectively. Earlier studies also suggested that fertilizers and diesel for machinery operations incurred more energy requirement than other inputs such as seed and labor [24,25,39]. In this study, land preparation and harvesting and threshing of rice needed more energy than *rabi* crops because puddling and transplanting operations required greater amounts of energy (Table 11).

**Table 11.** Amount of energy (MJ ha$^{-1}$) consumed in various rice-based cropping systems under sesbania green manuring and rice residue incorporation in rainfed lowlands.

| Input | *Sesbania* Green Manuring | | | | Rice Residue Incorporation | | | |
|---|---|---|---|---|---|---|---|---|
| | Rice Fallow | Rice Lentil | Rice Linseed | Rice Rapeseed | Rice Fallow | Rice Lentil | Rice Linseed | Rice Rapeseed |
| Seed of Green manure crop | 735 | 735 | 735 | 735 | | | | |
| Green manure matter/rice residue | 67,500 | 67,500 | 67,500 | 67,500 | 62,500 | 62,500 | 62,500 | 62,500 |
| Nitrogen (100% N) | 5291 | 5291 | 5291 | 5291 | 5291 | 5291 | 5291 | 5291 |
| Phosphorus | 498 | 498 | 498 | 498 | 498 | 498 | 498 | 498 |
| Potassium | 446 | 446 | 446 | 446 | 446 | 446 | 446 | 446 |
| Seed | 515 | 1103 | 1103 | 588 | 515 | 1103 | 1103 | 588 |
| Pesticides | 238 | 238 | 238 | 476 | 238 | 238 | 238 | 476 |
| Diesel | 1971 | 2647 | 2647 | 2647 | 1971 | 2647 | 2647 | 2647 |
| Tractor | 752 | 1129 | 1129 | 1129 | 752 | 1129 | 1129 | 1129 |
| Human labor | | | | | | | | |
| Men | 1646 | 2117 | 2180 | 2274 | 1646 | 2117 | 2180 | 2274 |
| Women | 1030 | 1281 | 1356 | 1382 | 1030 | 1281 | 1356 | 1382 |

The annual energy requirement of rice-based cropping systems mainly depends on crop management, which ranged from 70,720 MJ ha$^{-1}$ with rice residue incorporation to 76,793 MJ ha$^{-1}$ with *Sesbania* green manuring (Table 12). Even though *Sesbania* green manuring consumed more energy, it resulted in a higher energy output (182,657 MJ ha$^{-1}$), net energy (105,864 MJ ha$^{-1}$), energy intensity (1.68 MJ INR$^{-1}$) and human energy profitability (787) than the rice residue incorporation. However, rice residue incorporation recorded a higher energy ratio (2.42), energy productivity (0.082 kg MJ$^{-1}$) and energy profitability (1.42 kg MJ$^{-1}$) than the *Sesbania* green manuring, which was mainly due to less labor and machinery being required for rice-residue incorporation. The higher energy output under *Sesbania* green manuring was mainly due to the highest biological yield of *kharif* rice as well as *rabi* crops.

**Table 12.** Effect of green manuring, rice residue and fertility levels on input–output energy, net energy, energy ratio, energy-use efficiency, energy productivity and energy intensity under rainfed rice-based cropping system.

| Treatment | Input Energy | Output Energy MJ ha$^{-1}$ | Net Energy | Energy Ratio | Human Energy Profitability | Energy Profitability | Energy Productivity kg MJ$^{-1}$ | Energy Intensity MJ INR$^{-1}$ |
|---|---|---|---|---|---|---|---|---|
| **Rice residue management** | | | | | | | | |
| Green manuring | 76,793 | 182,657 | 105,864 | 2.38 | 787 | 1.38 | 0.080 | 1.68 |
| Rice residue incorporation | 70,720 | 171,292 | 100,572 | 2.42 | 758 | 1.42 | 0.082 | 1.49 |
| **Fertility levels** | | | | | | | | |
| Control | 70,250 | 157,972 | 87,722 | 2.25 | 690 | 1.25 | 0.077 | 1.56 |
| 50% RDF | 73,367 | 172,415 | 99,048 | 2.35 | 753 | 1.35 | 0.080 | 1.59 |
| 75% RDF | 74,926 | 185,035 | 110,110 | 2.47 | 808 | 1.47 | 0.083 | 1.58 |
| 100% RDF | 76,484 | 192,477 | 115,992 | 2.52 | 841 | 1.52 | 0.083 | 1.60 |
| **Succeeding Rabi crops** | | | | | | | | |
| Rice Fallow | 72,477 | 163,493 | 91,016 | 2.26 | 886 | 1.26 | 0.070 | 1.75 |
| Rice Lentil | 74,260 | 178,064 | 103,804 | 2.40 | 759 | 1.40 | 0.082 | 1.51 |
| Rice Linseed | 74,276 | 186,305 | 112,029 | 2.51 | 765 | 1.51 | 0.090 | 1.49 |
| Rice Rapeseed | 74,014 | 180,037 | 106,023 | 2.43 | 716 | 1.43 | 0.082 | 1.62 |

The input and output energy increased with increasing fertility levels from control to 100% RDF, but output energy per unit use of fertilizers started declining from 50% RDF up to 100% RDF. However, the maximum energy consumption (76,484 MJ ha$^{-1}$), output (192,477 MJ ha$^{-1}$), energy ratio, human energy profitability, energy profitability, energy productivity and energy intensity were with the highest level of fertility 100% RDF, which was mainly due to higher biomass as well as grain yield production of rice and succeeding crops with 100% RDF. The input energy requirement for the rice-fallow system was lower

(72,477 MJ ha$^{-1}$) as most of the inputs were used in the raising rice crop and no energy incurred while keeping land fallow. The rice-linseed cropping system required more input energy than the rice-lentil and rice-rapeseed cropping system, which was mainly due to more labor engaged in management. However, the rice-linseed cropping system resulted in a higher energy output (186,305 MJ ha$^{-1}$) and net energy (112,029 MJ ha$^{-1}$) than other systems. The rice-linseed cropping system also obtained higher values for energy ratio, energy profitability and productivity with the least energy intensity, which showed that the rice-linseed cropping system is the most efficient rice-based cropping system in rainfed lowland ecologies as under such situations linseed thrives better in terms of productivity than lentil and rapeseed in dry winters. Rice fallow recorded the lowest values for most of the energy parameters such as energy output, net energy, energy ratio, energy profitability and productivity with the highest energy intensity (1.75 MJ INR$^{-1}$) which showed that rice cultivation is highly energy intensive. Rice-based cropping systems are generally considered less energy efficient because of lower productivity [24,25,39] but inclusion of oilseeds and pulses efficiently utilized resources and enhanced productivity [35,36].

## 4. Conclusions

In the present scenario, there is an urgent need to identify a sustainable intensification option of rice fallows with efficient nutrient management protocols. Our 3-year study suggested that *Sesbania* green manuring coupled with 75% of recommended dose of fertilizers (RDF) is a more productive, resource-smart and sustainable nutrient management strategy for intensified rice systems of rainfed shallow lowlands of Eastern India. Intensification of rice fallows through incorporating short-duration pulses and oilseeds crops in rotation is relatively more profitable, energy efficient, water smart and sustainable than mono-cropping of rice. The findings of present study can be recommended in rice fallows of Eastern India which covers > 14 m ha area. Among the cropping systems studied, the rice-linseed system was the most productive and energy efficient, followed by rice rapeseed. Nevertheless, the experimentation was carried out with utmost care with the available set of resources, but there were certain limitations, a major one being a reduced number of cropping systems in the study. Therefore, in-depth studies under pragmatic on-farm environments for future work will assist in understanding sustainable intensification in more comprehensive way. Further, thorough understanding of nutrient dynamics in a system mode, long-term soil biological and chemical health effects, and development of location-specific moisture management protocols using low-cost bio-resources and polymers in the new intensified system can be a good future line of work.

**Author Contributions:** Conceptualization, T.S., B.S.S. and B.L.; data curation, R.S.B.; formal analysis, T.S. and R.S.B.; investigation, T.S., R.S.B. and B.S.S.; methodology, T.S., B.S.S. and B.L.; project administration, T.S. and B.S.S.; resources, T.S., B.S.S. and B.L.; software, R.S.B. and A.K.Y.; supervision, R.S., T.S. and B.S.S.; validation, R.S.B. and R.S.; writing—original draft, R.S.B., T.S. and A.K.Y.; writing—review and editing, R.S., T.S. and R.S.B. All authors have read and agreed to the published version of the manuscript.

**Funding:** This research received no external funding.

**Institutional Review Board Statement:** Not applicable.

**Informed Consent Statement:** Not applicable.

**Data Availability Statement:** Not applicable.

**Acknowledgments:** The authors are thankful to ICAR-National Rice Research Institute, Cuttack ICAR–Indian Agricultural Research Institute, New Delhi for providing necessary facilities during the conduct of the study and preparation of this manuscript.

**Conflicts of Interest:** The authors declare no conflict of interest.

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
