# Peer review of "Energy Balance, Productivity and Resource-Use Efficiency of Diverse Sustainable Intensification Options of Rainfed Lowland Rice Systems under Different Fertility Scenarios"

_sustainability, doi:10.3390/su14063657_

Round 1
Reviewer 1 Report
All of comments in attachment.

Author Response
Respected Sir/Madam,
Below are the response.
Regards,
Reviewer #1
Comment:
All of comments in attachment (Clarification of years in methodology and minor correction in figures)
Action: All corrections have been incorporated in the manuscript.
Reviewer 2 Report
Some suggestion for improving the paper are follows:
- The efficiency, and sustainability measures used should be described. The efficiency, productivity and sustainability status before and after implementing the study should be investigated and reported.
- Used international standard for expressing all measures used, see figure 1 symbols in expressing month.
- Novelty and limitation of the study should be expressed clearly.
- Managerial implication as well as recommendation should be expressed clearly.
Author Response
Respected Sir/Madam,
Please see the attachment.
Regards,
Authors

Reviewer 3 Report
Dear, Authors,
Soil fertility and sustainable crop yield is important issue in growing World population. The effective use of soil properties is an important issue in rice-fallow systems. Beside research actuality some comments on the article itself:
- How the ranges of Maximum and Minimum temperatures were determined? What led to such a temperatures range choice?
- Some Fig.1 X axis are incompletely shown. What are relations between three pictures of Fig. 1 X axis and titles 2013-2014, 2014-2015 and 2015-2016?
- Title of the Table 1 needs to be checked for writing orderliness.
- Check the text for lack of spaces between words.
- Line 143 what is a reason of the @ sign?
- Formulas in the text should contain the References if those are not created by authors.
- Line 248-251 - the text in different colour
Author Response

(The authors gave the same response as above.)

Reviewer 4 Report
This manuscript presents a study on energy balance, productivity and resource-use efficiency of diverse sustainable intensification options of rainfed-lowland rice systems under different fertility scenarios. Overall, the authors have made good effort to deliver their findings through this manuscript. However, I believed this manuscript needs enhancements in order to be better and meet the standard of this journal. My comments and suggestions are as follows:
- Abstract is not that concise to wrap-up the content of the study. Abstract shall contain a brief of the following elements e.g. introduction, method, results, and discussion, and conclusions. Some these important elements of the abstract in are not found in the current abstract.
2. Research gaps also are not that compact to articulate the actual research issues or research questions. I suggest the authors to re-write the research gap (L94-95) so that it would be concise and more comprehensible.
3. I suggest the climate data (fig.1) to be replaced with Tables. If the authors prefer use figures, thus, re-write some of texts, which are not written in English. ( The texts in X-axis are not written in English).
4. The authors must add the detailed specifications of the machinery/tools/equipment used in the experiment (L149-L153).
5. The author must write the citation for Energy Equivalent in Table 3. Because I believed the Energy Equivalent work came from another source.
Author Response

(The authors gave the same response as above.)

Reviewer 5 Report
There are values in the graphics in the text that are not readable. It could be necessary to better explain the passage from table 2 to 3 on energy conversion, considering it to be a crucial part of the work
Author Response
Respected Sir/Madam,
Below are the response.
Regards,
Author
Reviewer #5
Comment:
There are values in the graphics in the text that are not readable. It could be necessary to better explain the passage from table 2 to 3 on energy conversion, considering it to be a crucial part of the work.
Reply:
Corrected as per suggestion in fig.1. Details of Table 2 & 3 are given in the text in para 2.5.
Round 2
Reviewer 2 Report
The authors have addressed the reviewer commends and suggestions, therefor I have no objection the article be published.